# TRIP: Refining Image-to-Image Translation via Rival Preferences

## Abstract

We propose a new model to refine image-to-image translation via an adversarial ranking process. In particular, we simultaneously train two modules: a generator that translates an input image to the desired image with smooth subtle changes with respect to some specific attributes; and a ranker that ranks rival preferences consisting of the input image and the desired image. Rival preferences refer to the adversarial ranking process: (1) the ranker thinks no difference between the desired image and the input image in terms of the desired attributes; (2) the generator fools the ranker to believe that the desired image changes the attributes over the input image as desired. Preferences over pairs of real images are introduced to guide the ranker to rank image pairs regarding the interested attributes only. With an effective ranker, the generator would "win" the adversarial game by producing high-quality images that present desired changes over the attributes compared to the input image. The experiments demonstrate that our TRIP can generate high-fidelity images which exhibit smooth changes with the strength of the attributes.

## 1 Introduction

Image-to-image (I2I) translation (Isola et al., 2017) aims to translate an input image into the desired ones with changes in some specific attributes. Current literature can be classified into two categories: binary translation (Zhu et al., 2017; Kim et al., 2017), e.g., translating an image from "not smiling" to "smiling"; fine-grained translation (Lample et al., 2017; He et al., 2019; Liu et al., 2018; Saquil et al., 2018), e.g., generating a series of images with smooth changes from "not smiling" to "smiling". In this work, we focus on the high-quality fine-grained I2I translation, namely, generate a series of realistic versions of the input image with smooth changes in the specific attributes (See Fig. 1). Note that the desired high-quality images in our context are two folds: first, the generated images look as realistic as training images; second, the generated images are only modified in terms of the specific attributes.

Relative attribute (RA), referring to the preference of two images over the strength of the interested attribute, is widely used in the fine-grained I2I translation task due to their rich semantic information. Previous work Ranking Conditional Generative Adversarial Network (RCGAN) (Saquil et al., 2018) adopts two separate criteria for a high-quality fine-grained translation. Specifically, a ranker is adopted to distill the discrepancy from RAs regarding the targeted attribute, which then guides the generator to translate the input image into the desired one. Meanwhile, a discriminator ensures the generated images as realistic as the training images. However, the generated fine-grained images guided by the ranker are out of the real data distribution, which conflicts with the goal of the discriminator. Therefore, the generated images cannot maintain smooth changes and suffer from low-quality issues. RelGAN (Wu et al., 2019) applied a unified discriminator for the high-quality fine-grained translation. The discriminator guides the generator to learn the distribution of triplets, which consist of pairs of images and their corresponding numerical labels (i.e., relative attributes). Further, RelGAN adopted the fine-grained RAs within the same framework to enable a smooth interpolation. However, the joint data distribution matching does not explicitly model the discrepancy from the RAs and fails to capture sufficient semantic information. The generated images fail to change smoothly over the interested attribute.

In this paper, we propose a new adversarial ranking framework consisting of a ranker and a generator for high-quality fine-grained translation. In particular, the ranker explicitly learns to model the

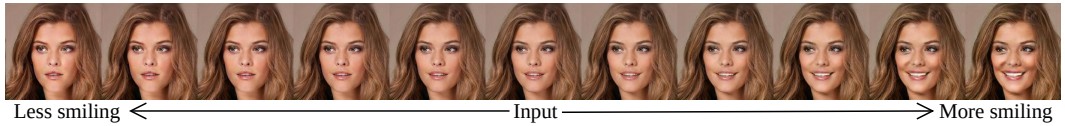

Less smiling ⟵———————————————— Input ————————————⟶ More smiling

Figure 1: Fine-grained Image-to-image translation on the "smile" attribute (generated by our TRIP).

discrepancy from the relative attributes, which can guide the generator to produce the desired image from the input image. Meanwhile, the rival preference consisting of the generated image and the input image is constructed to evoke the adversarial training between the ranker and the generator. Specifically, the ranker cannot differentiate the strength of the interested attribute between the generated image and the input image; while the generator aims to achieve the agreement from the ranker that the generated image holds the desired difference compared to the input. Competition between the ranker and the generator drives both two modules to improve themselves until the generations exhibit desired preferences while possessing high fidelity. We summarize our contributions as follows:

- We propose Translation via RIval Preference (TRIP) consisting of a ranker and a generator for a high-quality fine-grained translation. The rival preference is constructed to evoke the adversarial training between the ranker and the generator, which enhances the ability of the ranker and encourages a better generator.

- Our tailor-designed ranker enforces a continuous change between the generated image and the input image, which promotes a better fine-grained control over the interested attribute.

- Empirical results show that our TRIP achieves the state-of-art results on the fine-grained image-to-image translation task. Meanwhile, the input image can be manipulated linearly along the strength of the attribute.

- We further extend TRIP to the fine-grained I2I translation of multiple attributes. A case study demonstrates the efficacy of our TRIP in terms of disentangling multiple attributes and manipulating them simultaneously.

## 2 RELATED WORKS

We mainly review the literature related to fine-grained I2I translation, especially smooth facial attribute transfer. We summarized them based on the type of generative models used.

**AE/VAE-based** methods can provide a good latent representation of the input image. Some works (Lample et al., 2017; Liu et al., 2018; Li et al., 2020; Ding et al., 2020) proposed to disentangle the attribute-dependent latent variable from the image representation but resorted to different disentanglement strategies. Then the fine-grained translation can be derived by smoothly manipulating the attribute variable of the input image. However, the reconstruction loss, which is used to ensure the image quality, cannot guarantee a high fidelity of the hallucinated images.

**Flow-based** Some works (Kondo et al., 2019) incorporates feature disentanglement mechanism into flow-based generative models. However, the designed multi-scale disentanglement requires large computation. And the reported results did not show satisfactory performance on smooth control.

**GAN-based** GAN is a widely adopted framework for a high-quality image generation. Various methods applied GAN as a base for fine-grained I2I translation through relative attributes. The main differences lie in the strategies of incorporating the preference over the attributes into the image generation process. Saquil et al. (2018) adopted two critics consisting of a ranker, learning from the relative attributes, and a discriminator, ensuring the image quality. Then the combination of two critics is supposed to guide the generator to produce high-quality fine-grained images. However, the ranker would induce the generator to generate out-of-data-distribution images, which is opposite to the target of the discriminator, thereby resulting in poor-quality images. Wu et al. (2019) applied a unified discriminator, which learns the joint data distribution of the triplet constructed with a pair of images and a discrete numerical label (i.e., relative attribute). However, such a joint distribution modeling approach only models the discrete discrepancy of the RAs, which fails to generalize to the continuous labels very well. Rather than using RAs, He et al. (2019) directly modeled the attribute

Figure 2: The network structure of TRIP. ① and ② denotes different image pairs, i.e., real image pairs and generated image pairs, corresponding to Fig. 3 and Fig. 4, respectively. $R$ and $D$ denotes the rank head and the GAN head, respectively.

with binary classification, which cannot capture detailed attribute information, and hence fail to make a smooth control over the attributes. Deng et al. (2020) embedded 3D priors into adversarial learning. However, it relies on available priors for attributes, which limits the practicality. Alharbi and Wonka (2020) proposed an unsupervised disentanglement method. It injects the structure noises to GAN for controlling specific parts of the generated images, which makes global or local features changed in a disentangled way. However, it is unclear how global or local features are related to facial attributes. Thus, it is difficult to change specific attributes.

Our method is based on GAN. To ensure good control over the target attribute, the critic in GAN should transfer the signal about the subtle difference over the target attribute to the generator. Previous methods model it as two sequential processes. Namely, they capture the subtle difference over attribute using a classification model or a ranking model, and count on the learned attribute model to generalize learned attribute preference to the unseen generated images through interpolation. However, the learned attribute model never meets our expectation, since they haven't seen the generated images at all during its training. As for our TRIP, we consider introducing the generated image into the training process of the attribute model, i.e., the ranker. Since the supervision over the generated images is not accessible, we formulate the ranker into an adversarial ranking process using the constructed rival preference, following the adversarial training of vanilla GAN. Consequently, our ranker (the attribute model) can critic the generated image during its whole training process, and it no doubt can generalize to generated images to ensure sufficient fine-grained control over the target attribute.

## 3 TRIP FOR FINE-GRAINED IMAGE-TO-IMAGE TRANSLATION

In this section, we propose a new model, named TRanslation via Rival Preferences (TRIP) for high-quality fine-grained image-to-image (I2I) translation, which learns a mapping that translates an input image to a set of realistic output images by smoothly controlling the specific attributes.

The whole structure of TRIP is shown in Fig. 2, which consists of a generator and a ranker. The generator takes as input an image along with a continuous latent variable that controls the change of the attribute, and outputs the desired image; while the ranker provides information in terms of image quality and the preference over the attribute, which guides the learning of the generator. We implement the generator with a standard encoder-decoder architecture following Wu et al. (2019). In the following, we focus on describing the detailed design of the ranker and the principle behind it.

### 3.1 RANKER FOR RELATIVE ATTRIBUTES

Relative attributes (RAs) are assumed to be most representative and most valid to describe the information related to the relative emphasis of the attribute, owing to its simplicity and easy construction (Parikh and Grauman, 2011; Saquil et al., 2018). For a pair of images $(\mathbf{x}, \mathbf{y})$, RAs refer to their preference over the specific attribute: $\mathbf{y} \succ \mathbf{x}$ when $\mathbf{y}$ shows a greater strength than $\mathbf{x}$ on the target attribute and vice versa.

Pairwise learning to rank is a widely-adopted technique to model the relative attributes (Parikh and Grauman, 2011). Given a pair of images $(\mathbf{x}, \mathbf{y})$ and its relative attribute, the pairwise learning to rank

technique is formulated as a binary classification (Cao et al., 2006), i.e.,

$$R(\mathbf{x}, \mathbf{y}) = \begin{cases} 1 & \mathbf{y} \succ \mathbf{x}; \\ -1 & \mathbf{y} \prec \mathbf{x}, \end{cases} \tag{1}$$

where $R(\mathbf{x}, \mathbf{y})$ is the ranker's prediction for the pair of images $(\mathbf{x}, \mathbf{y})$.

Further, the attribute discrepancy between RAs, distilled by the ranker, can then be used to guide the generator to translate the input image into the desired one.

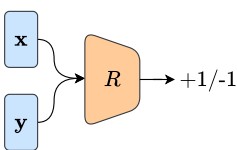

However, the ranker is trained on the real image pairs, which only focuses on the modeling of preference over the attribute and ignores image quality. To achieve the agreement with the ranker, the generator possibly produces unrealistic images, which conflicts with the goal of the discriminator.

Figure 3: The ranker model.

### 3.2 RIVAL PREFERENCES ENHANCING THE RANKER

According to the above analysis, we consider incorporating the generated image pairs into the modeling of RAs, along with the real image pairs to reconcile the goal of the ranker and the discriminator. Meanwhile, the resultant ranker will not only generalize well to the generated pairs but also avoid providing untrustworthy feedback by discriminating the unrealistic images.

Motivated by the adversarial training of GAN, we introduce an adversarial ranking process between a ranker and a generator to incorporate the generated pairs into the training of ranker. To be specific,

- **Ranker.** Inspired by semi-supervised GAN (Odena, 2016), we assign a pseudo label to the generated pairs. In order to avoid a biased influence on the ranking decision over real image pairs, i.e., positive (+1) or negative (-1), the pseudo label is designed as zero. Note that the generated pair consists of a synthetic image and its input in order to connect the ranker prediction to the controlling latent variable.

$$R(\mathbf{x}, \Delta) = \begin{cases} +1 & \Delta = \mathbf{y} \wedge \mathbf{y} \succ \mathbf{x}; \\ -1 & \Delta = \mathbf{y} \wedge \mathbf{y} \prec \mathbf{x}; \\ 0 & \Delta = \hat{\mathbf{y}}. \end{cases} \tag{2}$$

  where $\hat{\mathbf{y}}$ denotes the output of the generator given the input image $\mathbf{x}$ and $v$, i.e., $\hat{\mathbf{y}} = G(\mathbf{x}, v)$. $\Delta$ is a placeholder that can either be a real image $\mathbf{y}$ or be a generated image $\hat{\mathbf{y}}$.

- **Generator.** The goal of the generator is to achieve the consistency between the ranking prediction $R(\mathbf{x}, \hat{\mathbf{y}})$ and the corresponding latent variable $v$. When $v > 0$, the ranker is supposed to believe that the generated image $\hat{\mathbf{y}}$ has a larger strength of the specific attribute than the input $\mathbf{x}$, i.e., $R(\mathbf{x}, \hat{\mathbf{y}}) = +1$; and vice versa.

$$R(\mathbf{x}, \hat{\mathbf{y}}) = \begin{cases} +1 & v > 0; \\ -1 & v < 0. \end{cases} \tag{3}$$

We denominate the opposite goals between the ranker and the generator w.r.t. the generated pairs as **rival preferences**[1]. An intuitive example of the rival preference is given in Fig. 4 for better understanding.

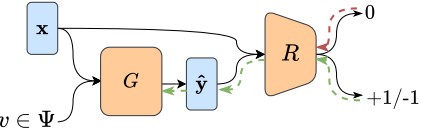

The ranker is promoted in terms of the following aspects: (1) The function of the ranker on the real image pairs is not changed. The generated pairs are uniformly sampled regarding their latent variables. By assigning label zero, the ranking information implied within the pairs is neutral-

Figure 4: Rival Preferences for the generated pairs between the ranker and the generator. $\Psi$ denotes $[-1, 0) \cup (0, 1]$.

ized to maintain the ranking performance on the real image pairs. (2) The ranker avoids providing biased ranking prediction for unrealistic image pairs. As we constrain the generated pairs at the decision boundary, i.e, $R(\mathbf{x}, \hat{\mathbf{y}}) = 0$, the ranker is invariant to the features specified by the generated

---

[1]"rival" means adversarial. We use it to distinguish it from adversarial training in the community.

pairs (Chapelle et al., 2008), suppressing the influence of the unrealistic features on the ranking decision. (3) The ranker can capture the exclusive difference over the specific attribute through the adversarial process. Since the ranker rejects to give effective feedback for unrealistic image pairs, only the realistic image pairs can attract the attention of the ranker. Therefore, the ranker only passes the effective information related to the target attribute to the generator.

Then, we introduce a parallel head following the feature layer to ensure the image quality together with a rank head, shown in Fig. 2. According to the above analysis, the ranker would not evoke conflict with the goal of the image quality. Therefore, we successfully reconcile the two goals of image quality and the extraction of the attribute difference. With a powerful ranker, the generator would "win" the adversarial game by producing the realistic pairs consistent with the latent variable.

**Remark 1** (Assigning zero to similar real image pairs)**.** *It is natural to assign zero to pairs* $\{(\mathbf{x}, \mathbf{y}) | \mathbf{y} = \mathbf{x}\}$*, where = denotes that* $\mathbf{x}$ *and* $\mathbf{y}$ *have same strength in the interested attribute. They can improve the ranking prediction (Zhou et al., 2008).*

### 3.3 Linearizing the Ranking Output

Equation 3 models the relative attributes of the generated pairs as a binary classification, which fails to enable a fine-grained translation since the subtle changes implied by the latent variable are not distinguished by the ranker. For example, given $v_1 > v_2 > 0$, the ranker give same feedbacks for $(\mathbf{x}, \hat{\mathbf{y}}_1)$ and $(\mathbf{x}, \hat{\mathbf{y}}_2)$ are both +1, which loses the discrimination between the two pairs. To achieve the fine-grained translation, we linearize the ranker's output for the generated pairs so as to align the ranker prediction with the latent variable. We thus reformulate the binary classification as the regression:

$$R(\mathbf{x}, \hat{\mathbf{y}}) = v. \tag{4}$$

Note that the output of the ranker can reflect the difference in a pair of images. Given two latent variables $1 > v_2 > v_1 > 0$, the ranking predictions for the pair generated from $v_2$ should be larger than that from $v_1$, i.e., $1 > R(\mathbf{x}, \hat{\mathbf{y}}_2) > R(\mathbf{x}, \hat{\mathbf{y}}_1) > 0$. The ranker's outputs for the generated pairs would be linearly correlated to the corresponding latent variable. Therefore, the generated output images can change smoothly over the input image according to the latent variable.

### 3.4 Translation via RIval Preferences (TRIP)

In the following, we introduce the loss functions for the two parallel heads in the ranker. The overall network structure can be seen in Fig. 2.

**Loss of rank head** $R$**:** we adopt the least square loss for the ranking predictions. The loss function for the ranker and the generator is defined as:

$$L_{rank}^{R} = \mathbb{E}_{p(\mathbf{x}, \mathbf{y}, r)} \left[ (R(\mathbf{x}, \mathbf{y}) - r)^2 \right] + \lambda \mathbb{E}_{p(\mathbf{x})p(v)} \left[ (R(\mathbf{x}, G(\mathbf{x}, v)) - 0)^2 \right]; \tag{5a}$$

$$L_{rank}^{G} = \mathbb{E}_{p(\mathbf{x})p(v)} \left[ (R(\mathbf{x}, G(\mathbf{x}, v)) - v)^2 \right], \tag{5b}$$

where $\hat{\mathbf{y}} = G(\mathbf{x}, v)$. $r = \begin{cases} 1 & \mathbf{y} \succ \mathbf{x} \\ -1 & \mathbf{y} \prec \mathbf{x} \end{cases}$ denotes the relative attribute. $p(\mathbf{x}, \mathbf{y}, r)$ are the joint distribution of real image preferences. $p(\mathbf{x})$ is the distribution of the training images. $p(v)$ is a uniform distribution $[-1, 1]$. $\lambda$ is the weight factor that considers the rival preferences.

By optimizing $L_{rank}^{D}$ (equation 5a), the ranker is trained to predict correct labels for real image pairs and assign zero for generated pairs, i.e., equation 2. By optimizing $L_{rank}^{G}$ (equation 5b), the generator is trained to output the desire image $\hat{\mathbf{y}}$, where the difference between $\hat{\mathbf{y}}$ and $\mathbf{x}$ is consistent with the latent variable $v$, i.e., equation 4.

**Loss of GAN head** $D$**:** to be consistent with the above rank head and also ensure a stable training, a regular least square GAN's loss is adopted:

$$L_{gan}^{D} = \mathbb{E}_{p(\mathbf{x})} \left[ (D(\mathbf{x}) - 1)^2 \right] + \mathbb{E}_{p(\mathbf{x})p(v)} \left[ (D(G(\mathbf{x}, v)) - 0)^2 \right]; \tag{6a}$$

$$L_{gan}^{G} = \mathbb{E}_{p(\mathbf{x})p(v)} \left[ (D(G(\mathbf{x}, v)) - 1)^2 \right], \tag{6b}$$

where $1$ denotes the real image label while $0$ denotes the fake image label.

Jointly training the rank head and the gan head, the gradients backpropagate through the shared feature layer to the generator. Thus our TRIP can conduct the high-quality fine-grained I2I translation.

### 3.5 Extended to the Multiple Attributes

To generalize our TRIP to multiple ($K$) attributes, we use vectors $\mathbf{v}$ and $\mathbf{r}$ with $K$ dimension to denote the latent variable and the preference label, respectively. Each dimension controls the change of one of the interested attributes. In particular, the ranker consists of one GAN head and $K$ parallel rank head. The overall loss function is summarized as follows:

$$L_{rank}^{R} = \mathbb{E}_{p(\mathbf{x},\mathbf{y},\mathbf{r})} \sum_{k} \left[ (R_k(\mathbf{x},\mathbf{y}) - \mathbf{r}_k)^2 \right] + \lambda \mathbb{E}_{p(\mathbf{x})p(\mathbf{v})} \sum_{k} \left[ (R_k(\mathbf{x}, G(\mathbf{x},v)) - 0)^2 \right]; \quad (7a)$$

$$L_{rank}^{G} = \mathbb{E}_{p(\mathbf{x})p(\mathbf{v})} \sum_{k} \left[ (R_k(\mathbf{x}, G(\mathbf{x},v)) - \mathbf{v}_k)^2 \right], \quad (7b)$$

where $R_k$ is the output of the $k$-th rank head. $\mathbf{v}_k$ and $\mathbf{r}_k$ are the $k$-th dimension of $\mathbf{v}$ and $\mathbf{r}$, respectively.

## 4 Experiments

In this section, we compare our TRIP with various baselines on the task of fine-grained image-to-image translation. We verify that our ranker can distinguish the subtle difference in a pair of images. Thus we propose to apply our ranker for evaluating the fine-graininess of image pairs generated by various methods. We finally extend TRIP to the translation of multiple attributes.

**Datasets.** We conduct experiments on the high quality version of a subset from Celeb Faces Attributes Dataset (CelebA-HQ) (Karras et al., 2018) and Labeled Faces in the Wild with attributes (LFWA) (Liu et al., 2015). CelebA-HQ consists of 30K face images of celebrities, annotated with $40$ binary attributes such as hair colors, gender and age. LFWA has $13,143$ images with $73$ annotated binary attributes. We resize the images of two datasets to $256 \times 256$. The relative attributes are obtained for any two images $\mathbf{x}$ and $\mathbf{y}$ only based on the binary label of attributes. For instance, for "smiling" attribute, we construct the comparison $\mathbf{x} > \mathbf{y}$ when the label of $\mathbf{x}$ is "smiling" while the label of $\mathbf{y}$ is "not smiling", and vice versa. Therefore, we make a fair comparison with other baselines in terms of same supervision information.

**Implementation Details.** As the translation is conducted on the unpaired setting, the cycle consistency loss $L_{cycle}$ (Zhu et al., 2017) are usually introduced to keep the identity of faces when translation. An orthogonal loss $L_o$ and the gradient penalty loss $L_{gp}$ are added to stabilize the training following (Wu et al., 2019). The weighting factor for $L_{gan}$, $L_{cycle}$, $L_o$ and $L_{gp}$ are $\lambda_g$, $\lambda_c$, $\lambda_o$ and $\lambda_{gp}$, respectively. Except $\lambda_g = 0.5$ for CelebA-HQ and $\lambda_g = 5$ for LFWA, we set the same parameter for all datasets. Specifically, we set $\lambda = 0.5$, $\lambda_c = 2.5, \lambda_{gp} = 150, \lambda_o = 10^{-6}$. We use the Adam optimizer [23] with $\beta_1 = 0.5$ and $\beta_2 = 0.999$. The learning rate is set to $1e$-5 for the ranker and $5e$-5 for the generator. The batch size is set to 4. See appendix for details about the network architecture and the experiment setting.

**Baselines.** We compare TRIP with FN (Lample et al., 2017), RelGAN (Wu et al., 2019) and RCGAN (Saquil et al., 2018). We use the released codes of FN, RelGAN and RCGAN [2]. We did not compare with AttGAN since RelGAN outperforms AttGAN, which is shown in (Wu et al., 2019).

**Evaluation Metrics.** Follow Wu et al. (2019), we use three metrics to quantitatively evaluate the performance of fine-grained translation. Standard deviation of structural similarity (SSIM) measures the fine-grained translation. Frechet Inception Distance (FID) measures the visual quality. Accuracy of Attribute Swapping (AAS) evaluates the accuracy of the binary image translation. The swapping for the attribute is to translate an image, e.g., from "smiling" into "not smiling". The calculation can be found in App. D.

---

[2]https://github.com/facebookresearch/FaderNetworks, https://github.com/willylulu/RelGAN, https://github.com/saquil/RankCGAN

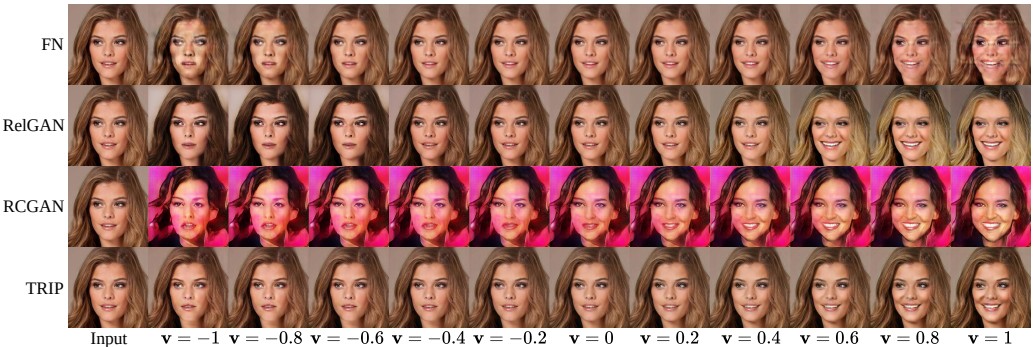

FN
RelGAN
RCGAN
TRIP

Input  $\mathbf{v}=-1$  $\mathbf{v}=-0.8$  $\mathbf{v}=-0.6$  $\mathbf{v}=-0.4$  $\mathbf{v}=-0.2$  $\mathbf{v}=0$  $\mathbf{v}=0.2$  $\mathbf{v}=0.4$  $\mathbf{v}=0.6$  $\mathbf{v}=0.8$  $\mathbf{v}=1$

Figure 5: Comparison of fine-grained facial attribute ("smile") translation on CelebA-HQ dataset.

| Model | Fine-grained (SSIM) | | | | | | Image quality (FID) | | | | | |
| | CelebA-HQ | | | | LFWA | | CelebA-HQ | | | | LFWA | |
| | Smile | Gender | Mouth | Cheekbones | Smiling | Frown | Smile | Gender | Mouth | Cheekbones | Smiling | Frown |
|---|---|---|---|---|---|---|---|---|---|---|---|---|
| FN | 0.0122 | **0.0036** | 0.0075 | 0.0039 | 0.0065 | **0.0005** | 41.48 | 48.66 | 42.79 | 43.15 | **12.58** | **11.37** |
| RCGAN | 0.0084 | 0.0138 | - | 0.0100 | 0.0079 | 0.0106 | 398.05 | 418.06 | - | 385.27 | 437.93 | 425.59 |
| RelGAN | 0.0512 | 0.0924 | 0.0261 | 0.0510 | 0.0137 | 0.016 | 10.92 | 31.29 | 9.55 | 10.28 | 16.76 | 16.21 |
| TRIP | **0.0030** | 0.0077 | **0.0017** | **0.0028** | **0.0001** | **0.0005** | **10.19** | **26.47** | **7.18** | **9.50** | 22.25 | 23.65 |

Table 1: Fine-grained performance (SSIM) and image quality (FID) of FN, RCGAN, RelGAN and TRIP on CelebA-HQ and LFWA. RCGAN fails to make a fine-grained translation w.r.t. the "mouth" attribute on CelebA-HQ dataset. So we did not collect its result.

## 4.1 FINE-GRAINED IMAGE-TO-IMAGE TRANSLATION

We conduct fine-grained I2I translation on a single attribute. On CelebA-HQ dataset, we make the I2I translation in terms of "smile", "gender", "mouth open" and "high cheekbones" attributes, respectively. On LFWA dataset, we make the I2I translation in terms of "smile" and "Frown" attributes, respectively. We show that our TRIP achieves the best performance on the fine-grained I2I translation task comparing with various strong baselines.

**Best visual results** As shown in Fig. 5, (1) all GANs can translate the input image into "more smiling" when $\mathbf{v} > 0$ or "less smiling" when $\mathbf{v} < 0$. The degree of changes is consistent with the numerical value of $\mathbf{v}$. (2) Our GAN's generation shows the best visual quality, generating realistic output images that are different from the input images only in the specific attribute. In contrast, FN suffers from image distortion issues. RelGAN's generation not only changed the specific attribute "smile", but also other irrelevant attributes, "hair color". RCGAN exhibits extremely poor generation results.

**Best fine-grained score** We present the quantitative evaluation of the fine-grained translation in Table 1. Our TRIP achieves the lowest SSIM scores, consistent with the visual results. Note that a trivial case to obtain a low SSIM is when the translation is failed. Namely, the generator would output the same value no matter what the latent variable is. Therefore, we further apply AAS to evaluate the I2I translation in a binary manner. Most GANs achieve over 95% accuracy except for FN (See the App. Fig. 12). Under this condition, it guarantees that a low SSIM indeed indicates the output images change smoothly with the latent variable.

**Best image quality score** Table 1 presents the quantitative evaluation of the image quality. (1) Our TRIP achieves the best image quality with the lowest FID scores. (2) FN achieves the best FID on LFWA dataset. Because the FN achieves a relatively low accuracy of the translation, $< 75\%$ in Fig. 12, many generated images would be the same as the input image. It means that the statistics of the translated images are similar to that of the input images, leading to a low FID. (3) RCGAN has the worst FID scores, consistent with the visual results in Fig. 1.

## 4.2 PHYSICAL MEANING OF RANKER OUTPUT

From Fig. 5 and Table 1, it shows that when conditioning on different latent variables, our TRIP can translate an input image into a series of output images that exhibit the corresponding changes over

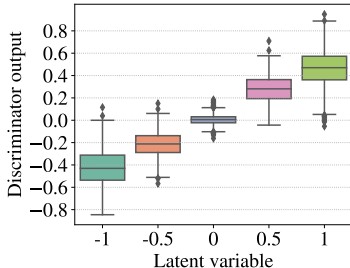

| $v$ | RelGAN | RCGAN | FN | TRIP |
|---|---|---|---|---|
| $-1.$ | 0.165 | 0.268 | 0.256 | **0.157** |
| $-0.5$ | 0.164 | 0.265 | 0.137 | **0.107** |
| $0.$ | **0.030** | 0.354 | 0.089 | 0.042 |
| $0.5$ | 0.183 | 0.273 | 0.122 | **0.121** |
| $1.$ | 0.172 | 0.290 | 0.222 | **0.159** |

Figure 7: The first three subfigures plot the ranker's output for generated pairs in terms of different latent variables. The curve shows the mean of the output, while the shaded region depicts one standard deviation of the output. We summarize the standard deviation in the table for better understanding.

the attribute. We then evaluate the function of our ranker using these fine-grained generated pairs. It verifies that our ranker's output well-aligns to the relative change in the pair of images.

We further evaluate fine-grained I2I translations w.r.t. the "smile" attribute on the test dataset of CelebA-HQ (See Fig. 6). The trained generator is applied to generate a set of $G(\mathbf{x}, v)$ by taking as inputs an image from the test dataset and $v = -1.0, -0.5, 0.0, 0.5, 1.0$, respectively[3]. Then we collect the output of the ranker for each generated pair and plot the density in terms of different types of pairs, i.e., with different $v$.

As shown in Fig. 6, (1) for a large $v$, the ranker would output a large prediction. It demonstrates that the ranker indeed generalizes to synthetic imaged pairs and can discriminate the subtle change among each image pair. (2) The ranker can capture the whole ordering instead of the exact value w.r.t. the latent variable. Because the ranker that assigns 0 to the generated pairs inhibits the generator's loss optimizing to zero, although our generator's objective is to ensure the ranker output values are consistent with the latent variable. However, the adversarial training would help the ranker to achieve an equilibrium with the generator when convergence, so that the ranker can maintain the whole ordering regarding the latent variable.

Figure 6: The box plot of the ranker's output for generated pairs with different values of the latent variable.

## 4.3 LINEAR TENDENCY ON THE LATENT VARIABLE

As our ranker can reveal the relative changes in pairs of images, we use it to evaluate the subtle differences of the fine-grained synthetic image pairs generated by different methods.

We generate the fine-grained pairs on the test dataset of CelebA-HQ w.r.t. the "smile" attribute. Each trained model produces a series of synthetic images by taking as input a real image and different latent variables. The range of the latent variable is from -1 to 1 with step 0.1[3]. Then the ranker, pre-trained by our TRIP, is applied to evaluate the generated pairs and group them in terms of different conditioned latent variables for different models, respectively. In terms of each group, we calculate the mean and the standard deviation (std) for the outputs of the ranker (Fig. 7).

Fig. 7 shows that (1) the ranking output of our TRIP exhibits a linear trend with the lowest variance w.r.t. the latent variable. This demonstrates that TRIP can smoothly translate the input image to the desired image over the specific attribute along the latent variable. (2) The ranking output of RCGAN behaves like a `tanh` curve with a sudden change when the latent variable is around zero. It means that RCGAN cannot smoothly control the attribute strength for the input image. In addition, RCGAN has the largest variance on the ranking output due to the low quality of the generated images, which introduces noises to the ranker's prediction on the generated pairs. (3) RelGAN manifests a three-step like curve, which indicates a failure of fine-grained generation. This is mainly because of its specific design of the interpolation loss. (4) FN presents a linear tendency like our TRIP, which denotes that it can make a fine-grained control over the attribute. However, the mean of the ranking output for the generated pairs is relatively low in FN, since it fails to translate some input images into the desired

---

[3]When conditioning on negative values of the latent variable, we use the test samples with the "smiling" attribute. When conditioning on positive values, we use the test samples with the "not smiling" attribute.

output images. This is verified by its low translation accuracy (See the appendix Fig. 12), lower than $85\%$. In addition, FN also exhibits a large variance of the ranking output due to the poor image quality.

## 4.4 EXTENDED TO MULTIPLE ATTRIBUTES

We conduct fine-grained I2I translation with two attributes "smile" and "male" on CelebA-HQ to show that our model can generalize well to the case with multiple attributes. We use the latent variable with two dimensions to control the change of "smile" and "male" attributes, respectively.

We show the generated outputs conditioning on different $\mathbf{v}$ in Fig. 8. (1) Our GAN can disentangle multiple attributes. When conditioning on $\mathbf{v} = [-1, 0]/[1, 0]$, the output images $O_{-1,0}/O_{1,0}$ appear "less smiling"/"more smiling" with no change in the "masculine" attribute.

When conditioning on $\mathbf{v} = [0, -1]/[0, 1]$, the output images $O_{0, -1}/O_{0,1}$ appear "less masculine"/"more masculine" with no change in the "smiling" attribute. In addition, a fine-grained control over the strength of a single attribute is still practical. (2) Our TRIP can manipulates the subtle changes of multiple attributes simultaneously. For example, when conditioning $\mathbf{v} = [1, -1]$, the output image $O_{1, -1}$ appear "less smiling" and "more masculine". Our TRIP can make a fine-grained translation on "smille" and "masculine".

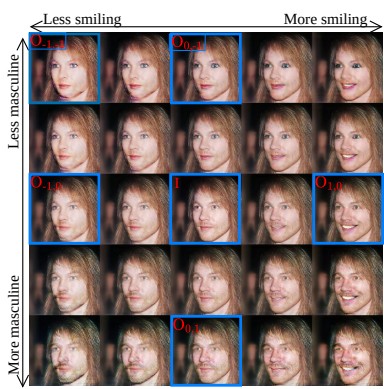

Figure 8: Fine-grained Image-to-image translation with "smile" and male attributes. The middle is the input image. Others are the generated output images conditioned on different $\mathbf{v}$.

## 5 CONCLUSION

In this paper, we propose a novel GAN for fine-grained image-to-image translation by modeling RAs, where the generated data is to model fake data region for the ranking model. We empirically show the efficacy of our GAN for the fine-grained translation on CelebA-HQ and LFWA dataset. Our proposed GAN can be deemed as a new form of semi-supervised GAN. The supervised pairwise ranking and the unsupervised generation target is incorporated into a single model function. So one of the promise of this work can be extended to semi-supervised GAN area.

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

| $L_{rank}$ | $L_{gan}$ | $\lambda$ | (3)/(4) | Results |
|---|---|---|---|---|
| | $\checkmark$ | 0.5 | (4) |  |
| $\checkmark$ | | 0.5 | (4) |  |
| $\checkmark$ | $\checkmark$ | 0 | (4) |  |
| $\checkmark$ | $\checkmark$ | 0.5 | (3) |  |
| $\checkmark$ | $\checkmark$ | 0.5 | (4) |  |

Figure 9: Ablation study. The experiments are the fine-grained Image-to-image translation on the "smile" attribute of CelebA-HQ. In the results, the first column is the input image. The other columns are the generated images conditioned on the latent variable from $-1$ to $1$ with step 0.2. (3) and (4) refers to equation 3 and equation 4, respectively. $L_{rank}$ refers to equation 5. $L_{gan}$ refers to equation 6.

| Model | AAS | SSIM | FID |
|---|---|---|---|
| $L_{rank} = 0$ | 9.73 | 1.51E-05 | 29 |
| $L_{gan} = 0$ | 98.47 | 0.0058 | 78.08 |
| $\lambda = 0$ | 55.7 | 0.0011 | 8.55 |
| CLS (3) | 92.37 | 0.0163 | 11.33 |
| TRIP | **97.69** | **0.0030** | **10.19** |

Table 2: Ablation study. Translation accuracy, (AAS), Fine-grained performance (SSIM) and image quality (FID) are reported, corresponding to the results in Fig. 12.

## A    ABLATION STUDY

In Fig. 9 and Table 2, we show an ablation study of our model. (1) Without $L_{rank}$, the generated images exhibit no change over the input image. The generator fails to learn the mapping of the translation, which is reflected by a extremely low translation accuracy. (2) Without $L_{gan}$, the image quality degrades, achieving a high FID score. (3) Setting $\lambda = 0$, i.e., without considering the adversarial ranking, the performance of facial image manipulation collapses, obtaining a low translation accuracy. (4) When optimizing with equation 3, i.e., not linearing the ranking output for the generated pairs, the fine-grained control over the attributes fails, getting a high SSIM score. (5) With our TRIP, the generated images present desired changes consistent with the latent variable and possess good quality.

## B    CONVERGENCE OF TRIP

We plot the training curve of the ranker and the generator, respectively, as shown in Fig. 10. It demonstrates that the ranker and the generator are trained against each other until convergence.

We plot the distribution of the ranker's prediction for real image pairs and generated image pairs with different relative attributes (RAs) $(+1/0/-1)$ using the ranker in Fig. 11. (1) At the beginning of the training (Fig. 11a), the ranker gives similar predictions for real image pairs with different RAs. The same observations can also be found on the generated image pairs. (2) After 100 iterations (Fig. 11b), the ranker learns to give the desired prediction for different kinds of pair, i.e., $> 0$ (averaged) for pairs with RA $(+1)$, 0 for pairs with RA $(0)$ and $-1$ for pairs with RA $(-1)$. (3) After $9,900$ iterations (Fig. 11c), TRIP converges. In terms of the real image pairs, the ranker output $+1$ for the pairs with RA $(+1)$, 0 for the pairs with RA $(0)$ and $-1$ for the pairs with RA $(-1)$ in the sense of average.

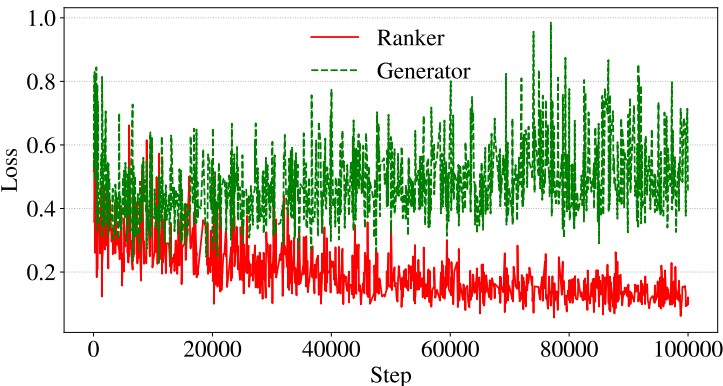

Figure 10: The curve of training loss. The ranker and the generator are trained against each other until convergence. Ranker loss $= L_{rank}^R + \lambda_g L_{gan}^R + \lambda_{gp} L_{gp}$. Generator loss $= L_{rank}^G + \lambda_g L_{gan}^G + \lambda_{gp} L_{cycle}$.

**This verifies that our ranker can give precise ranking predictions for real image pairs.** In terms of the generated pairs, the ranker outputs $+0.5$ for the pairs with RA $(+1)$, $0$ for the pairs with RA $(0)$ and $-0.5$ for the pairs with RA $(-1)$ in the sense of average. **This is a convergence state due to rival preferences.** We take pairs with RA $(+1)$ as an example. The generated pairs with RA $(+1)$ are expected to be assigned $0$ when optimizing the ranker and to be assigned $+1$ when optimizing the generator. Therefore, the convergence state should be around $0.5$. And so forth. This can explain why the ranker would output $0$ for the pairs with RA $(0)$ and $-0.5$ for pairs with RA $(-1)$ in the sense of average.

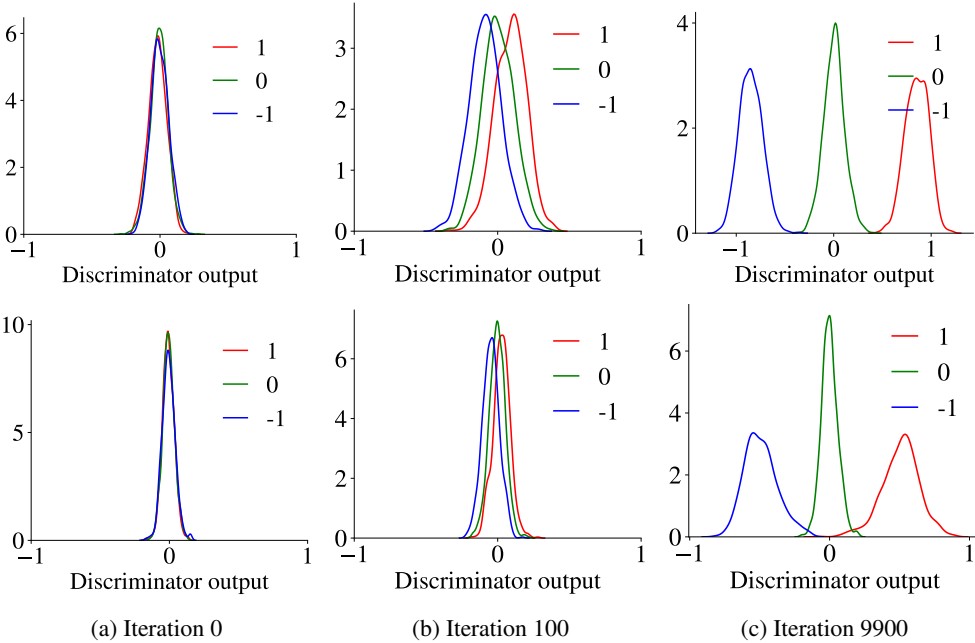

(a) Iteration 0      (b) Iteration 100      (c) Iteration 9900

Figure 11: The density plot of the discriminator's output for pairs of image from real dataset (first row) and generation of TRIP (second row) with different relative attributes $(+1/0/-1)$, respectively, during the training process.

## C    EXPERIMENTAL SETTING

### C.1    NETWORK STRUCTURE

Our generator will take one image and a random sampled relative attribute as input and output a translated image. Our generator network is same as RelGAN (Wu et al., 2019), which is composed of three convolutional layers for down-sampling, six residual blocks, and three transposed convolutional layers for up-sampling (shown in Table 3). Our proposed ranker will take pair of images $(\mathbf{x}, \mathbf{y})$ as inputs and output the classification score. It is comprised of two functional components rank layer and GAN layer following a feature layer (shown in Table 4).The rank layer and the GAN layer is for calculating $L_{rank}^{D}$ and $L_{gan}^{D}$, respectively. The feature layer $F$ is composed of six convolutional layers. The rank layer is composed of a subtract layer, one convolutional layer, one flatten layer and one dense layer. The subtract layer operates on $F(\mathbf{x})$ and $F(\mathbf{y})$, i.e., $F(\mathbf{y}) - F(\mathbf{x})$. The GAN layer is composed of one flatten layer and one dense layer.

### C.2    TRAINING

We split the dataset into training/test with a ratio $90/10$. We pretrain our GAN only with $L_{gan}$ to enable a good reconstruction for the generator. By doing so, we ease the training by sequencing the learning of our GAN. That is, we first make a generation with good quality. Then when our GAN begins to train, the ranker can mainly focus on the relationship between the generated pairs and its corresponding conditional $v$, rather than handling the translation quality and the generation quality together. All the experiment results are obtained by a single run.

| Component | Input → Output Shape | Layer Information |
|---|---|---|
| Down-sampling | $(h, w, 3 + K) \rightarrow (h, w, 64)$ | Conv-($N_f$=64,$S_f$=7,$S_s$=1,$S_p$=3),SN,ReLU |
| | $(h, w, 64) \rightarrow \left(\frac{h}{2}, \frac{w}{2}, 128\right)$ | Conv-($N_f$=128,$S_f$=4,$S_s$=2,$S_p$=1),SN,ReLU |
| | $\left(\frac{h}{2}, \frac{w}{2}, 128\right) \rightarrow \left(\frac{h}{4}, \frac{w}{4}, 256\right)$ | Conv-($N_f$=256,$S_f$=4,$S_s$=2,$S_p$=1),SN,ReLU |
| Residual Blocks | $\left(\frac{h}{4}, \frac{w}{4}, 256\right) \rightarrow \left(\frac{h}{4}, \frac{w}{4}, 256\right)$ | Residual Block: Conv-($N_f$=256,$S_f$=3,$S_s$=1,$S_p$=1),SN,ReLU |
| | $\left(\frac{h}{4}, \frac{w}{4}, 256\right) \rightarrow \left(\frac{h}{4}, \frac{w}{4}, 256\right)$ | Residual Block: Conv-($N_f$=256,$S_f$=3,$S_s$=1,$S_p$=1),SN,ReLU |
| | $\left(\frac{h}{4}, \frac{w}{4}, 256\right) \rightarrow \left(\frac{h}{4}, \frac{w}{4}, 256\right)$ | Residual Block: Conv-($N_f$=256,$S_f$=3,$S_s$=1,$S_p$=1),SN,ReLU |
| | $\left(\frac{h}{4}, \frac{w}{4}, 256\right) \rightarrow \left(\frac{h}{4}, \frac{w}{4}, 256\right)$ | Residual Block: Conv-($N_f$=256,$S_f$=3,$S_s$=1,$S_p$=1),SN,ReLU |
| | $\left(\frac{h}{4}, \frac{w}{4}, 256\right) \rightarrow \left(\frac{h}{4}, \frac{w}{4}, 256\right)$ | Residual Block: Conv-($N_f$=256,$S_f$=3,$S_s$=1,$S_p$=1),SN,ReLU |
| | $\left(\frac{h}{4}, \frac{w}{4}, 256\right) \rightarrow \left(\frac{h}{4}, \frac{w}{4}, 256\right)$ | Residual Block: Conv-($N_f$=256,$S_f$=3,$S_s$=1,$S_p$=1),SN,ReLU |
| Up-sampling | $\left(\frac{h}{4}, \frac{w}{4}, 256\right) \rightarrow \left(\frac{h}{2}, \frac{w}{2}, 128\right)$ | Conv-($N_f$=128,$S_f$=4,$S_s$=2,$S_p$=1),SN,ReLU |
| | $\left(\frac{h}{2}, \frac{w}{2}, 128\right) \rightarrow (h, w, 64)$ | Conv-($N_f$=64,$S_f$=4,$S_s$=2,$S_p$=1),SN,ReLU |
| | $(h, w, 64) \rightarrow (h, w, 3)$ | Conv-($N_f$=3,$S_f$=7,$S_s$=1,$S_p$=3),Tanh |

Table 3: Generator network architecture. We use switchable normalization, denoted as SN, in all layers except the last output layer. $N_f$ is the number of filters. $S_f$ is the filter size. $S_s$ is the stride size. $S_p$ is the padding size.

## D    EVALUATION

**SSIM.** We first apply the generator to produce a set of fine-grained output images $\{\mathbf{x}_1, \ldots, \mathbf{x}_{11}\}$ by conditioning an input image and a set of latent variable values from $-1$ to $1$ with a step $0.2$. We then compute **the standard deviation of the structural similarity (SSIM)** (Wang et al., 2004) between $x_{i-1}$ and $x_i$ as follows:

$$\sigma\left(\{\text{SSIM}\left(\mathbf{x}_{i-1}, \mathbf{x}_i\right) \mid i = 1, \cdots, 11\}\right). \tag{8}$$

We calculate SSIM for each image from the test dataset and average them to get the final score.

**AAS.** The accuracy is evaluated by a facial attribute classifier that uses the Resnet-18 architecture (He et al., 2016). To obtain AAS, we first translate the test images with the trained GANs and then apply the classifier to evaluate the classification accuracy of the translated images coupled with its swapping attribute. Higher accuracy means that more images are translated as desired.

**FID.** It is evaluated with $30K$ translated images on CelebA-HQ dataset and $13,143$ translated images on LFWA dataset.

# E    MORE EXPERIMENT RESULTS

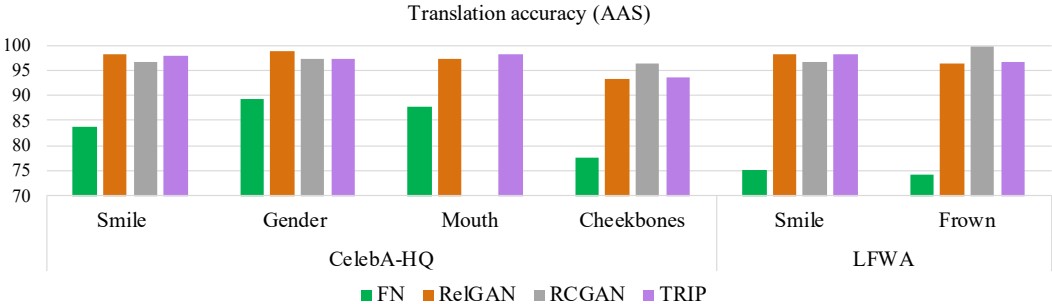

Figure 12: Translation accuracy (AAS, higher is better) of FN, RCGAN, RelGAN and TRIP on CelebA-HQ and LFWA. RCGAN fails to make a fine-grained translation w.r.t. the "mouth" attribute on CelebA-HQ dataset. So we did not collect its result.

# F    FINE-GRAINED I2I TRANSLATION ON NON-FACIAL DATASET

To further evaluate the effectiveness of our method, we conduct fine-grained I2I translation on shoes → edges from the UT Zappos50K dataset Isola et al. (2017). Fig. 16 shows that our TRIP can make a fine-grained translation from shoe images to edge images.

| Component | Input → Output Shape | Layer Information |
|---|---|---|
| Feature Layer | $2 \times (h, w, 3) \to 2 \times \left(\frac{h}{2}, \frac{w}{2}, 64\right)$ | Conv-($N_f$=64,$S_f$=4,$S_s$=2,$S_p$=1),LReLU |
| | $2 \times \left(\frac{h}{2}, \frac{w}{2}, 64\right) \to 2 \times \left(\frac{h}{4}, \frac{w}{4}, 128\right)$ | Conv-($N_f$=128,$S_f$=4,$S_s$=2,$S_p$=1),LReLU |
| | $2 \times \left(\frac{h}{4}, \frac{w}{4}, 128\right) \to 2 \times \left(\frac{h}{8}, \frac{w}{8}, 256\right)$ | Conv-($N_f$=256,$S_f$=4,$S_s$=2,$S_p$=1),LReLU |
| | $2 \times \left(\frac{h}{8}, \frac{w}{8}, 256\right) \to 2 \times \left(\frac{h}{16}, \frac{w}{16}, 512\right)$ | Conv-($N_f$=512,$S_f$=4,$S_s$=2,$S_p$=1),LReLU |
| | $2 \times \left(\frac{h}{16}, \frac{w}{16}, 512\right) \to 2 \times \left(\frac{h}{32}, \frac{w}{32}, 1024\right)$ | Conv-($N_f$=1024,$S_f$=4,$S_s$=2,$S_p$=1),LReLU |
| | $2 \times \left(\frac{h}{32}, \frac{w}{32}, 1024\right) \to 2 \times \left(\frac{h}{64}, \frac{w}{64}, 2048\right)$ | Conv-($N_f$=2048,$S_f$=4,$S_s$=2,$S_p$=1),LReLU |
| Rank Layer | $2 \times \left(\frac{h}{64}, \frac{w}{64}, 2048\right) \to \left(\frac{h}{64}, \frac{w}{64}, 2048\right)$ | Subtract |
| | $\left(\frac{h}{64}, \frac{w}{64}, 2048\right) \to \left(\frac{h}{64}, \frac{w}{64}, 2048\right)$ | Conv-($N_f$=1,$S_f$=1,$S_s$=1,$S_p$=1),LReLU |
| | $\left(\frac{h}{64}, \frac{w}{64}, 2048\right) \to \left(\frac{h}{64} \times \frac{w}{64} \times 2048\right)$ | Flatten |
| | $\left(\frac{h}{64} \times \frac{w}{64} \times 2048\right) \to (K, )$ | Dense |
| GAN Layer | $\left(\frac{h}{64}, \frac{w}{64}, 2048\right) \to \left(\frac{h}{64} \times \frac{w}{64} \times 2048\right)$ | Flatten |
| | $\left(\frac{h}{64} \times \frac{w}{64} \times 2048\right) \to (1, )$ | Dense |

Table 4: Ranker network architecture. LReLU is Leaky ReLU with a negative slop of 0.01. $K$ is the number of attributes. $N_f$ is the number of filters. $S_f$ is the filter size. $S_s$ is the stride size. $S_p$ is the padding size.

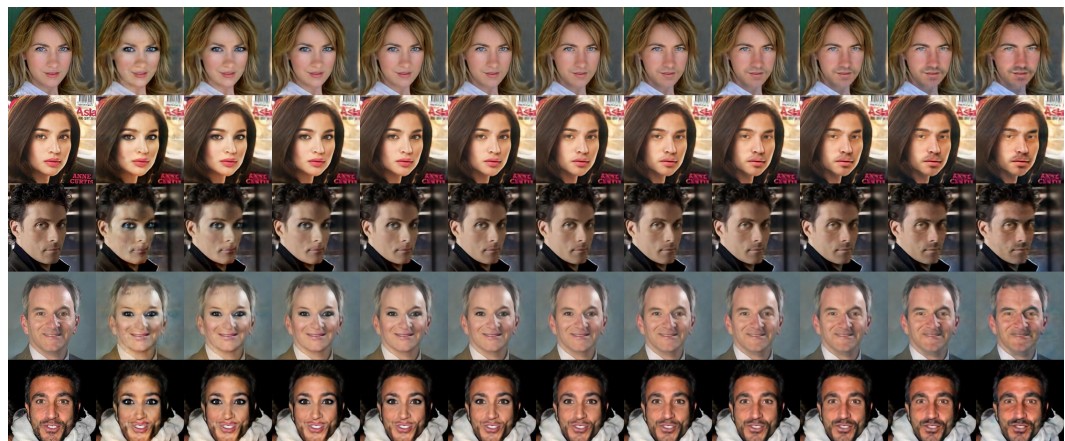

(a) FN

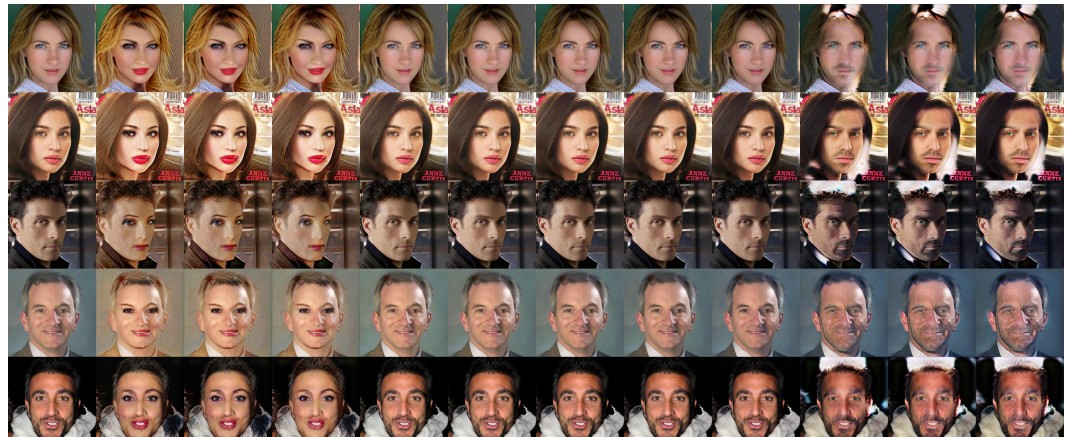

(b) RelGAN

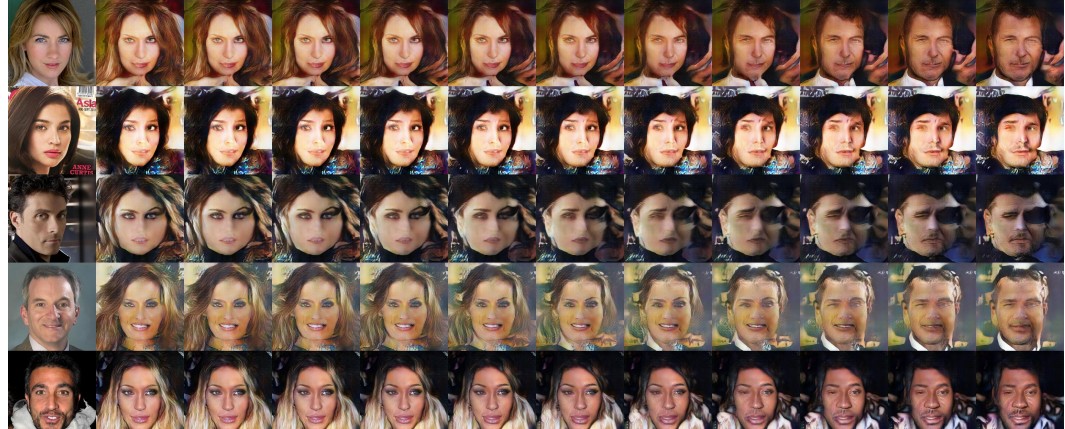

(c) RCGAN

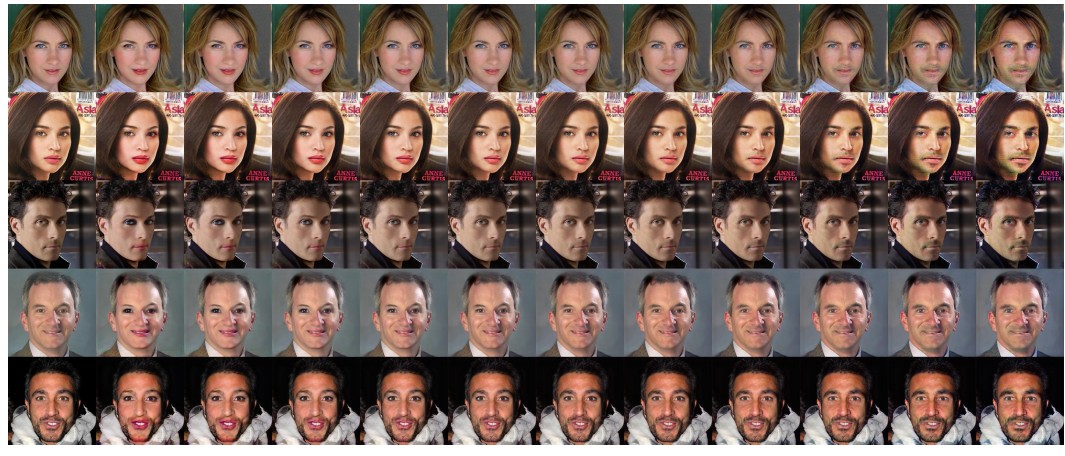

(d) TRIP

Figure 13: The fine-grained facial attribute ("male") translation on CelebA-HQ dataset applying various methods. The first column is the input image. The other columns are the generated images conditioned on the latent variable from −1 to 1 with step 0.2. The presented results are randomly sampled from the test set.

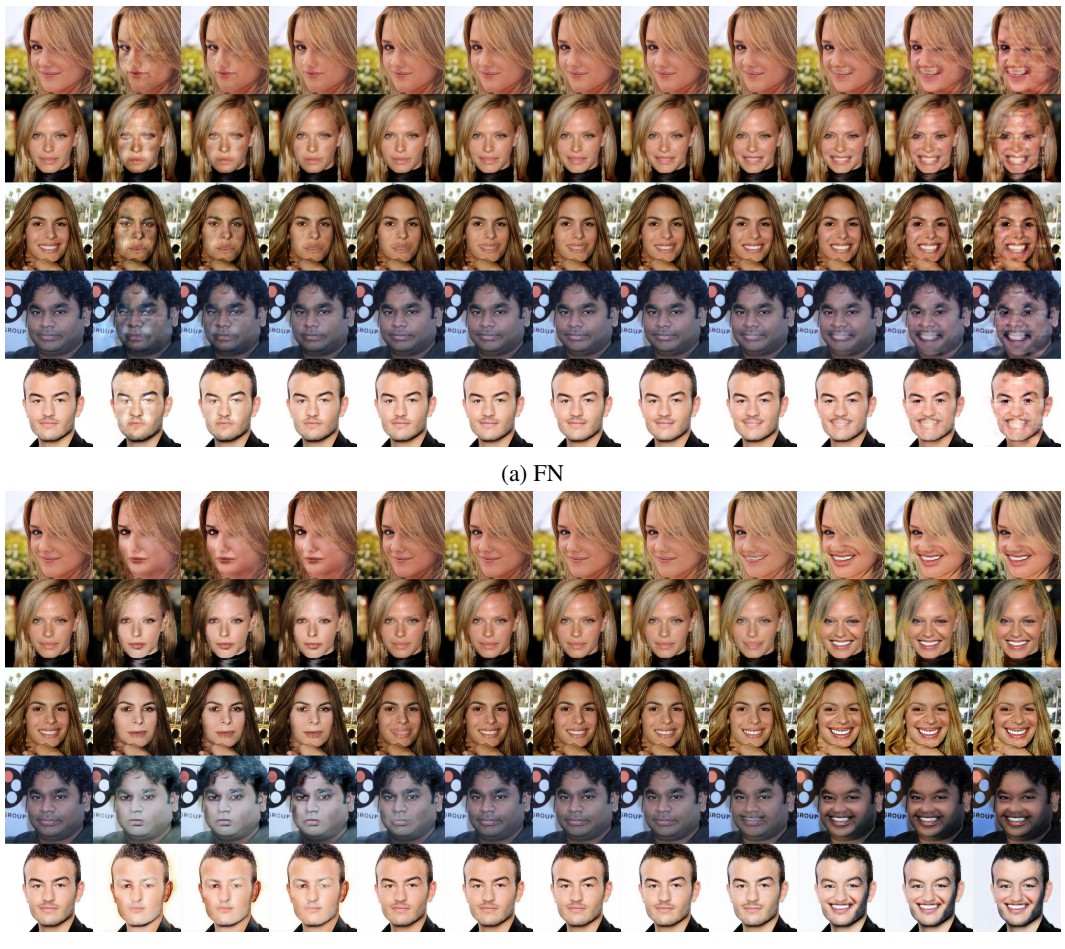

(a) FN

(b) RelGAN

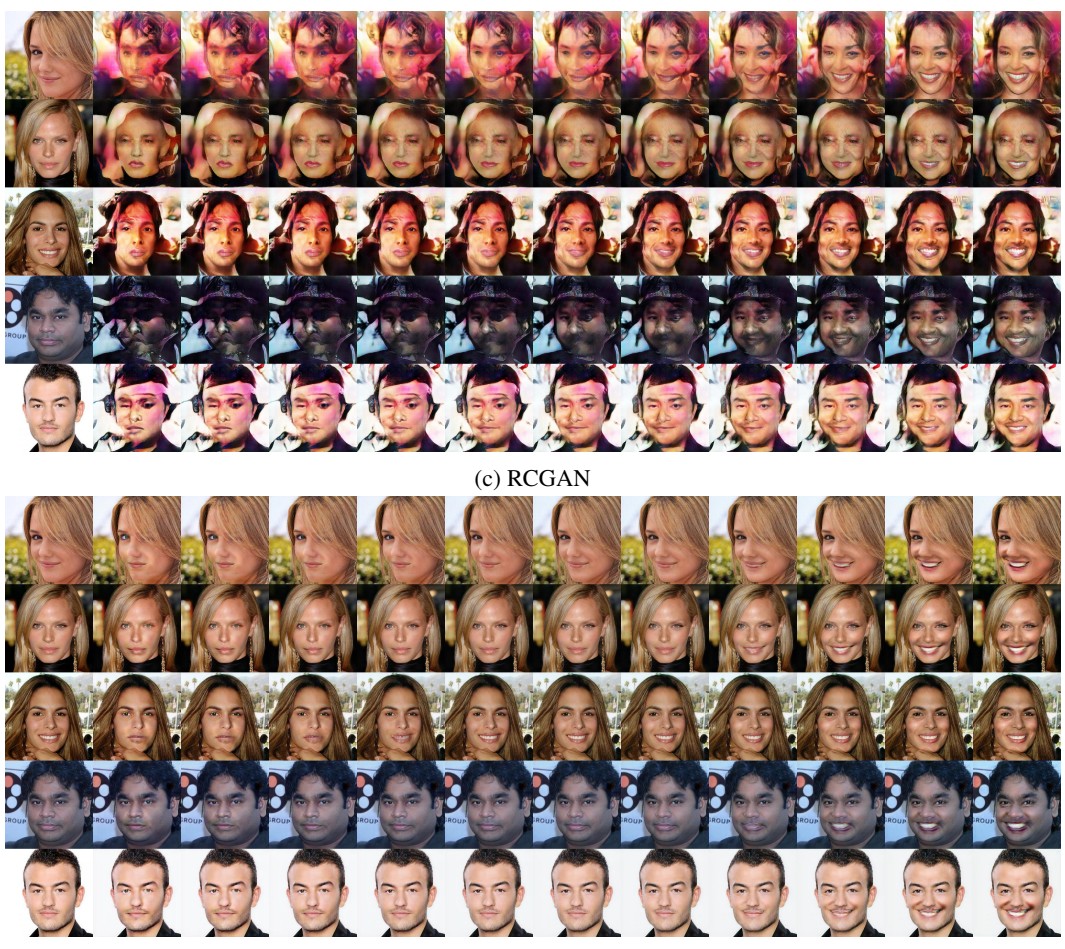

(c) RCGAN

(d) TRIP

Figure 14: The fine-grained facial attribute ("male") translation on CelebA-HQ dataset applying various methods. The first column is the input image. The other columns are the generated images conditioned on the latent variable from −1 to 1 with step 0.2. The presented results are randomly sampled from the test set.

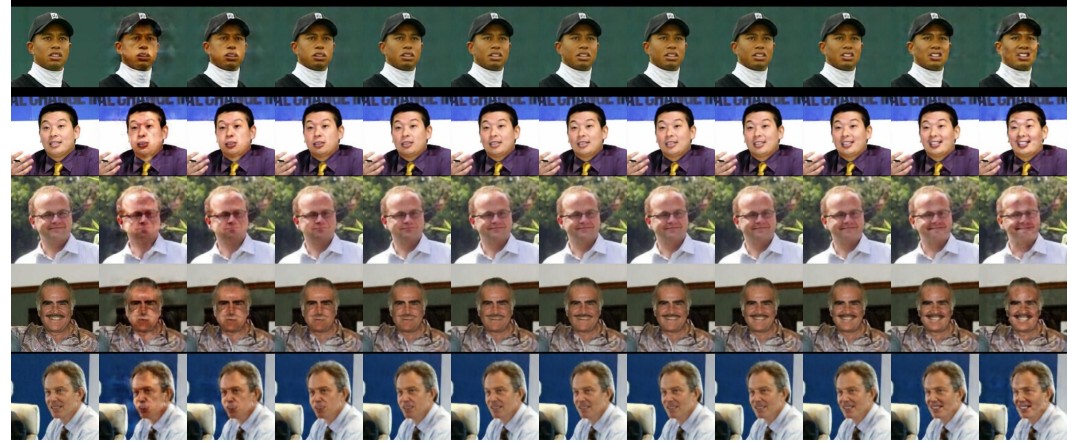

(a) FN

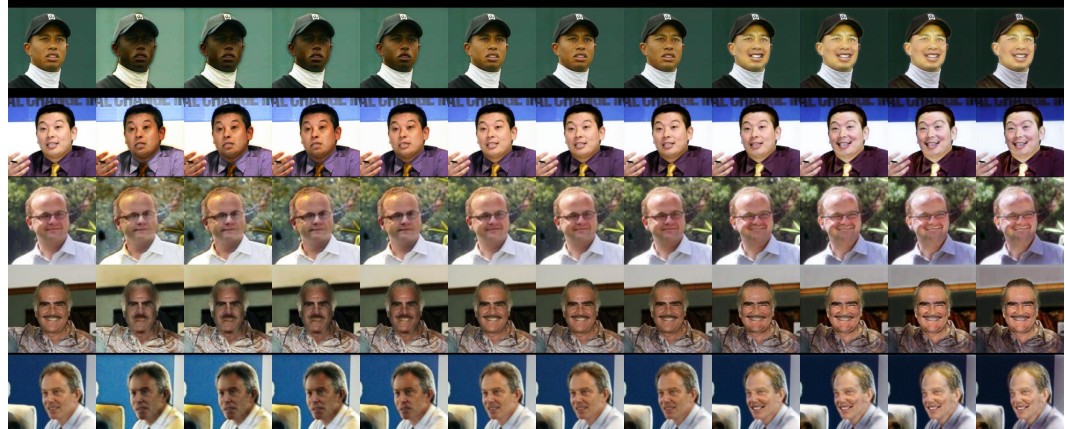

(b) RelGAN

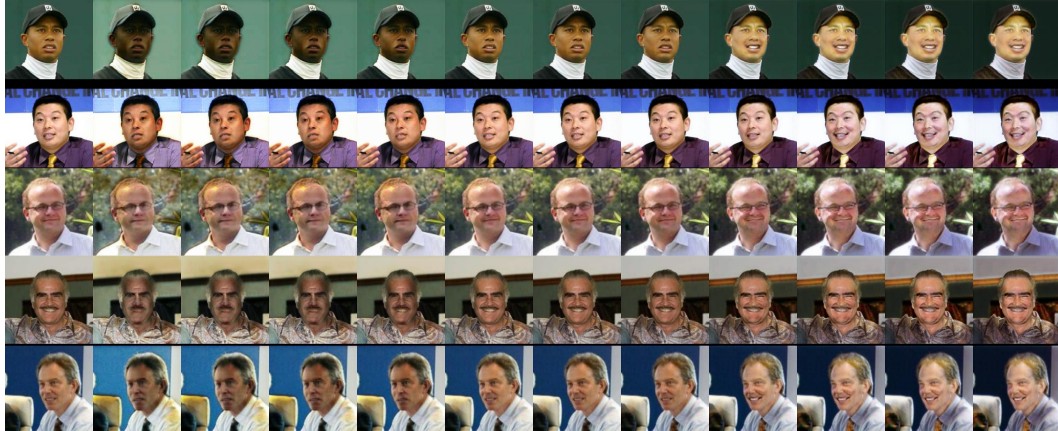

(c) RCGAN

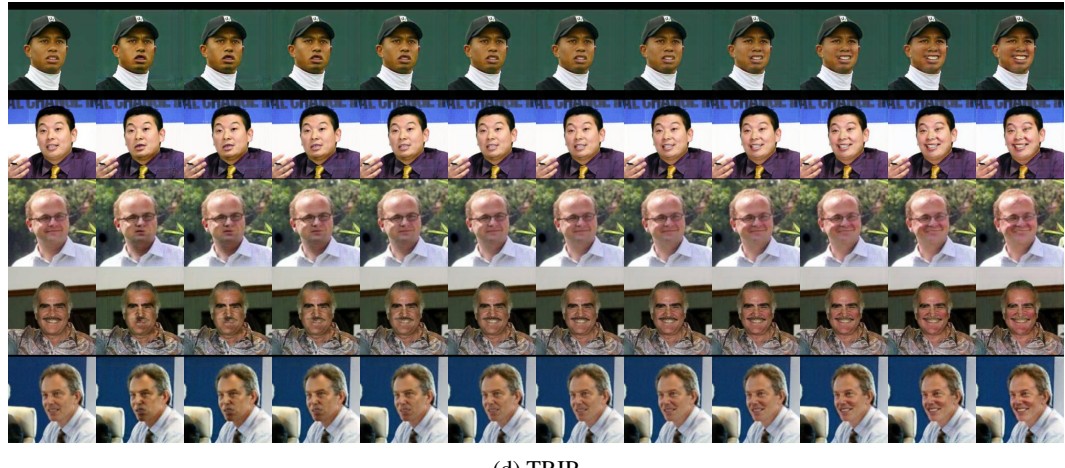

(d) TRIP

Figure 15: The fine-grained facial attribute ("male") translation on CelebA-HQ dataset applying various methods. The first column is the input image. The other columns are the generated images conditioned on the latent variable from $-1$ to $1$ with step 0.2. The presented results are randomly sampled from the test set.

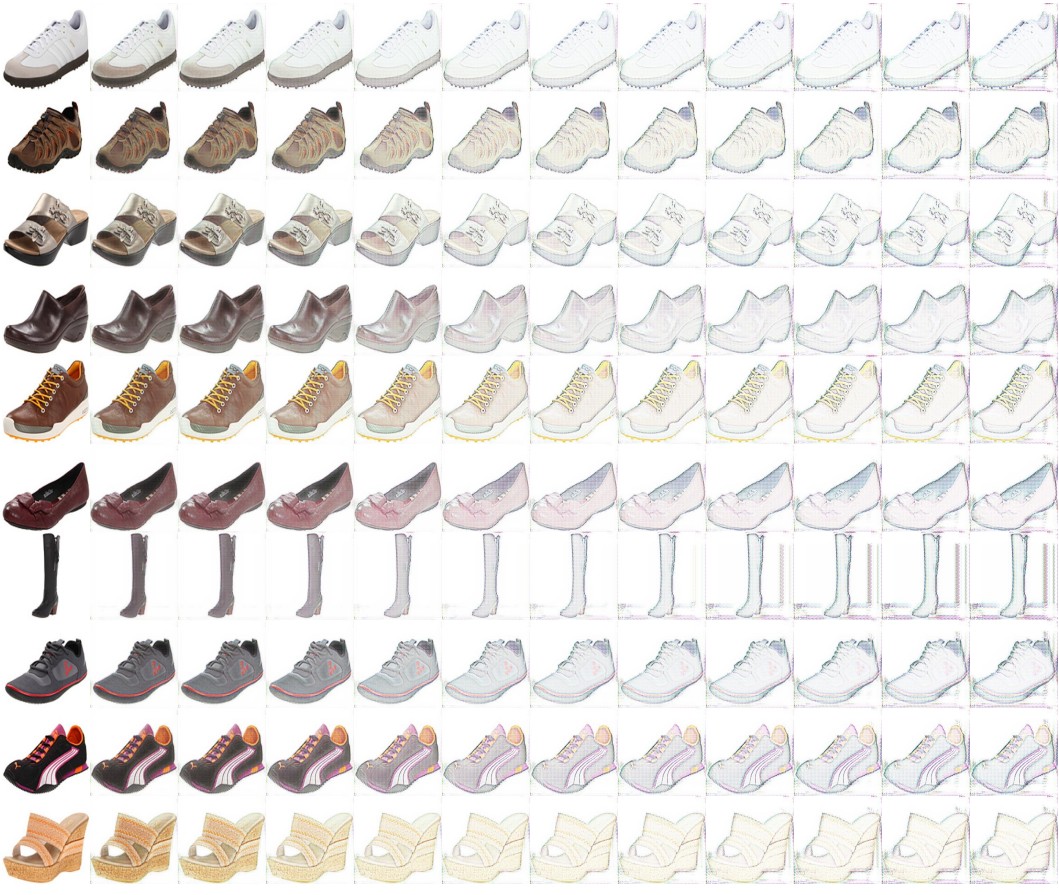

Figure 16: Fine-grained Image-to-image translation on the shoes $\rightarrow$ edges (generated by our TRIP)

