# OpenReview forum: "TRIP: Refining Image-to-Image Translation via Rival Preferences"
_ICLR.cc/2021/Conference — Reject_

### Official Review · AnonReviewer1 · 2020-10-27
**Interesting but limited contribution**

**Rating:** 4
**Confidence:** 4

**Review:**

[Summary]
This paper proposes a facial image manipulation method that incorporates ranker module using relative attributes and pairwise learning to rank, i.e. TRIP. The authors modify a typical discriminator of GAN into ranker with additional ranking head. They evaluate TRIP on CelebA-HQ and LFWA.

[Strength]
- Image manipulation is an interesting and practical topic.

[Weakness]
- Above all, I am not sure that how large can this work contributes. First, the authors argue that they address image-to-image (I2I) translation but the problem addressed is just facial image manipulation rather than I2I translation. In general, I2I term covers more wide topics: multidomain multimodal translation, style transfer, real-to-anime, etc. Second, this method was evaluated on facial image manipulation only. If the method can be applied to other domains (such as cars, birds, etc..), the contribution can be enhanced.
- Even if the authors claim relative attribute-based methods are major, I cannot agree with this argument. Recent many facial image manipulation work uses other methods (e.g. reference images[Choi et al. 2020], structured noise [Alharbi & Wonka 2020], semantic mask [Lee et al. 2020]) for high-fidelity images, showing promising results. Also, the author need to explicitly argue the main advantages of the proposed method in related work.
- Core related work were missed such as [Deng et al. 2020, Yan et al. 2019] except for the papers mention above.
- It is better that Figure captions are self-contained to improve the readability.
- The method part is not easy to follow. Some terms are used without definition (>, v). Eq(2) is not intuitive to understand. L^D should be L^R in 6(a).
- Even if TRIP shows better results than RelGAN in terms of quantitative metrics, the covered attributes of TRIP presented in the result section looks smaller than those of RelGAN.
- Considering all qualitative and quantitative results, I am not convinced that TRIP are superior than the baseline models.

[Ref]
- Choi et al. StarGAN v2: Diverse Image Synthesis for Multiple Domains. CVPR 2020.
- Deng et al. Disentangled and Controllable Face Image Generation via 3D Imitative-Contrastive Learning. CVPR 2020.
- Alharbi & Wonka Disentangled Image Generation Through Structured Noise Injection. CVPR2020.
- Lee et al. MaskGAN: Towards Diverse and Interactive Facial Image Manipulation. CVPR2020.
- Yan et al. Joint Deep Learning of Facial Expression Synthesis and Recognition. IEEE T on Multimedia 2019.

---

> ### Author Response · Authors · 2020-11-16
> **The first work to formulate facial attribute transfer as an adversarial ranking problem, achieving the SOTA performance**
>
> Thank you very much for your comments. Below is a report on how we address your concern.
>
> Weakness 1: Concerns about the contribution and the target scope of this work.
>    - R1: We follow the community [1,2] to denote the problem we addressed with the “image-to-image translation” term. This is the first work to formulate fine-grained I2I translation as an adversarial ranking problem, which achieves state-of-art results.
> Our TRIP also works on style transfer. We added the experiments of style transfer on the UT Zappos50K dataset [3]. It achieves good performance in translating shoe images to edge images in a fine-grained manner.
>
> Weakness 2 \& 3:	Concerns about the related methods (reference images[Choi et al. 2020], structured noise [Alharbi \& Wonka 2020], semantic mask [Lee et al. 2020]) and core related work were missed such as [Deng et al. 2020, Yan et al. 2019].
>    - R2 \& 3: We have rewritten the related work so as to correctly position our work into various facial image manipulation works. Note that it is manually cost to obtain the fine-grained reference images and semantic mask. We did not see any related work to use them for fine-grained translation. For the structure noise injection, it is unclear how global or local features are related to facial attributes. Thus, it is difficult to change specific attributes.
>
> Weakness 4 \& 5: concerns about figure captions and the writing of the method part.
>    - R4 \& 5: Thanks for the suggestion. We have revised it accordingly.
>
> Weakness 6:  the covered attributes of TRIP presented in the result section looks smaller than those of RelGAN.
>    - R6: Please note that RelGAN mainly targets multi-domain problems, i.e., manipulating multiple attributes simultaneously. For the fine-grained control over the specific attribute, it only covers three attributes, which is smaller than ours.
>
> Weakness 7: Considering all qualitative and quantitative results, I am not convinced that TRIP are superior than the baseline models.
>    - R7: We added more visual comparisons and more ablation studies to support our method's effectiveness and superiority than the baselines.
>
> Rerference
>    * [1] Choi, Yunjey, et al. "Stargan: Unified generative adversarial networks for multi-domain image-to-image translation." CVPR. 2018.
>    * [2] Wu, Po-Wei, et al. "Relgan: Multi-domain image-to-image translation via relative attributes." CVPR. 2019.
>    * [3] Zhu, J. Y., Park, T., Isola, P., & Efros, A. A.. Unpaired image-to-image translation using cycle-consistent adversarial networks. ICCV. 2017

---

### Official Review · AnonReviewer4 · 2020-10-28
**This paper deals with image manipulation though relative effect of a given attribute using a generator and a ranker.**

**Rating:** 4
**Confidence:** 4

**Review:**

Summary

The authors proposed in this paper a supervised approach relying on given relative and quantitative attribute discrepancies. A UNet-like generator learns adversarially tends to generate realistic images while a "ranker" tends to predict the magnitude of the input parameter used to control the image manipulation. The controled parameter is defined implictly using images whose the discrepancy of the attribute of interest is known. This allows fine-grained manipulation of the attribute of interest. The results of the approach is illustrated on face datasets (CelebA-HQ and LFWA).

Reason for score

The general idea behind the proposed approach is interesting and somehow novel in this context, but the positioning of the paper is a bit confusining and unfair according to ranker-based approaches. On top of that the quality of the writting (clarity, misleading notations…), the organization of the model section especially and the provided (and missing) details make it hard to follow (see below). Overall I vote for rejecting this paper.

Pros
1. Ranker-based approaches that require less supervision is essential to open image manipulation/translation to wider domain where its hard or costly to get precise labels while its generally most easier to relative information. But the proposed approach seem to be spuriously attached to this setting (see Cons 1.).
2. The idea of controlling changes in images through relative information is good and effective and it worth it to diffuse this knowledge in the field. For instance, it can simplify the problem (especially the particular case of no modification boils down to having the generator behaving like the identity function when the condition is null).
3.  Presented qualitative results are better compared to competition but I was wondering if the setting or the choosen example is fair enough (see question 2.)


Cons
1. Since relative attributes are obtained from real absolute attributes, the approach is more supervised (at least in the presented experiments) that it seems at first glance. Indeed, in actual relative supervision scheme, the datasets are more "quantitative" and generally fairly binary (like in pairwise comparisons or with triplets). For instance, we can expect that the differences from Fig.7 (more linear trend of the ranker) is only due to this fact. If not, it has to be demonstrated. So to be more fair on the potential and the interest of the approach, the supervision signal should be downgraded, or the authors should assume an higher degree of supervision and compare their results to standard (non-ranked based) conditioning based on absolute values of parameters.
2. Section 3.2: Since generated images have pseudo-label of 0, I don't understand why there passing through the ranker model during training rather than just being ignored (and removed from the rank loss). Besides, this seems to contradict the definition of $L_{rank}^G$ in equation 5b which tends to bring together $R(x,\hat{y})$ to $v$. My belief is the confusion comes from the fact you introduce first a standard binary ranker and then "linearized" the output. The section 3 should be rewritten and reorganised for the sake of clarity.
3. Since the whole optimized loss for D and G (and how there are trained: one after another or altogether ?) are not specified, it's not clear why $R(x,\hat{y})$ is present both in equation 5a and 5b. My understanding is loss 5a is used when optimizing D and loss 5b when optimizing G, which may lead to some unconsistancies and does not allow to cleary understand which mix-max game is optimized by this process. Besides additional loss are lost in the experiments section (cycle, gradient penalty…).
4. The authors claim that their approach can reach high-fidelity but all the proposed results still present strong artefacts even at a low resolution, even if better than competition, but high-fidelity may be too strong to recent work on image manipulation/translation.


Questions
1. Since the role of the "downsampling" part of the GAN is partly to encode the input image (+ the condition), why not have shared a common encoding module for "downsampling" and "Feature Layer" (a siamese network architecture could be used to handle the 3 inputs in this case) ? It would certainly help to reduce the number of weights and ease the training.
2. Can you clarify why your approach seems to only modify the targeted attribute (like in Fig. 5) compared to dAN which seem to derivate in your experimentations (whereas perfectly working examples are shown in RelGAN paper) ? Can you precise which component / principle could explain the difference (apart experimental biases or limitations) ?


Major comments
1. There is a lot of redundancies between the introduction and the "related works" section (especially on RCGAN and RelGAN). This could be better organized for clarity and to save important place for additional material. Additionnaly, the related work section essentially focus on rank-based approach (which is good to position the paper thinly, but not enough to position it more broadly), but is a bit scarce on other technics. For instance, VAE or flow-based approaches could be cited in this section, plus any approach that seek to structure and control changes directly from manipulation of the latent representation.
2. Fig2. vs Fig.3 are a bit misleading since the ranker model is acting on only 2 images while the "rank head" is taking 4. It is not well aligned with further explaination (for instance loss rank definition in section 3.4 that should be better reflected in Fig.2 and directly connected to $x$, $y$ and the output of the generator $\hat{y}$.
3. section 3.4 and 3.5: $Q$ notation does not really simplify equations but rather gives rise to some obfuscation. It's hard to understand your loss is only trying to bring together R(x,y) the prediction to r(x,y) the real order. Then, in equation 7a and 7b, $L_2$ norms could be used to simplify and clarify the equations.
4. In equation 5a, it does not seem possible to sample on $r$, since in fact, it depends on $x$ and $y$ and should be written $r(x,y)$ for clarity.


Minor comments
1. The "rival" wording used here and there from the abstract of the paper is not really introduced and self-supported.
2. In abstract: "real image preferences" was not clear for me before I see $y$ in Fig.2.
3. English: "evoke" verb is used several times in the paper, but I guess the intended meaning by the authors was not the more common understood definition. You may use: "provoke", "induce", "produce", "raise" according to the context.
4. In section 3.1, explaining how RCGAN uses its ranker its a bit confusing and should be better explained in the related work section.
5. section 3.3: closing parenthesis missing 3 lines after equation 4.
6. In equation 5a: notation error: it should be $L_{rank}^D$ rather than $L_{rank}^R$
7. Table 1: "UGGAN" name is mentionned rather than TRIP
8. section 4.1: typo: "best FID On" --> "best FID on"

---

> ### Author Response · Authors · 2020-11-16
> **Concerns about the fair comparison and our definition of high-quality (part 1/2)**
>
> Thank you very much for your comments. Below is a report for how we address your concern.
>
> Con1: concerns about the fair comparison and confusion of the linear trend of our method.
>    - Response: Sorry for the misleading statement about the construction of relative attributes. We have updated it in the revision. In our experiments, we obtain the relative attributes only based on the binary label of attributes. For instance,  for "smiling" attribute, we construct the comparison $\mathbf{x}>\mathbf{y}$ when the label of $\mathbf{x}$ is "smiling" while the label of $\mathbf{y}$ is not "smiling" and vice versa. Therefore, we make a fair comparison with other baselines in terms of the same supervision information.
>
>    The linear trend of our method does not result from using the relative attributes. RCGAN actually applied relative attributes also but fails to keep a linear trend as shown in Fig. 7. Our model contributes to this, which proposed the adversarial ranking process between the ranker and the generator as well as the tailor-designed ranking regression. We conduct ablation studies to support the claim, which is shown in the appendix.
>
> Cons 2: confusion about Section 3.2.
>    - Response: We pass the ranking loss for the generated pairs through the ranker model in order to make the ranker assign pseudo label zero for the generated pairs. The ranker then learns to distinguish real image pairs from generated pairs by assigning different labels from the real pairs', thus fusing the ranker's goal and the discriminator's goal into a unified ranker. The principle behind it is the same as semi-supervised GAN. The contradiction of the loss for the generator (equation 5b) and the loss for the ranker (equation 5a) is tailor-designed to evoke the adversarial training between the ranker and the generator, which enhances the ranker and then promotes a better generator.
>
> Cons 3: confusion about the training paradigm of our method and the additional loss.
>    - Response: Our training paradigm is exactly the same as the original GAN. The ranker and the generator are alternately trained until convergence. The defined loss is inherited from least-square GAN's principle [2], which also defines an adversarial game between the generator and the discriminator. The additional loss (gradient penalty, cycle..) are commonly used in GANs for I2I translation in order to stabilize the training or adapt to the task, like keeping face identity.
>
> Cons 4: The authors claim that their approach can reach high-fidelity but all the proposed results still present strong artifacts even at a low resolution, even if better than the competition, but high-fidelity may be too strong to recent work on image manipulation/translation.
>    - Response: We clarify our contribution in terms of image quality improvement more clearly in the revision. The desired high-quality images in our context are two folds: first, the generated images look as realistic as training images; second, the generated images are only modified in terms of the specific attributes. Our method achieves higher-quality images generation than all the baselines. Especially, our results are superior over current ranking-based methods, i.e., RCGAN (Fig. 5 \& Table 1).
>
>    We modified the related sentence in the main paper accordingly to avoid confusion.
>
> Q1: Why not have shared a common encoding module for "downsampling" and "Feature Layer" (a siamese network architecture could be used to handle the 3 inputs in this case)?
>    - Response: The feature layer of the ranker only encodes an image while the generator encodes the concatenation of an image and a conditional attribute. Since the inputs are different, the feature layer and the downsampling cannot be shared.
>
> Q2:	Can you clarify why your approach seems to only modify the targeted attribute compared to dAN which seem to derivate in your experimentations (whereas perfectly working examples are shown in RelGAN paper)? Can you precise which component/principle could explain the difference?
>    - Response: The reason why our TRIP only modifies the targeted attribute is two folds:
>       * (1) TRIP explicitly models the relative attributes via the ranker while RelGAN implicitly models the relative attributes by joint distribution matching.
>       * (2) the adversarial ranking process incorporates the generated images into the whole training process of the ranker, then the ranker can guide the generator to focus on the targeted attribute only.  While other baselines, like RelGAN, count on the generalization of the learned ranker model to the unseen generated images via interpolation, thus easily suffering from the contamination with other attributes.
>
>    Actually, the reported results in the original RelGAN also justify our claim. In Fig. 4 of the original RelGAN paper, the only attribute changed is supposed to be hair color, but significant changes in skin color, eyebrows, eyes, and lips, which is also reported in the paper [3].

---

> > ### Author Response · Authors · 2020-11-16
> > **Reply to the major commments and the minor comments (part 2/2)**
> >
> > Major C1: redundancies between the introduction and the "related works" section (especially on RCGAN and RelGAN).  Additionally, the related work section essentially focus on rank-based approach, but is a bit scarce on other technics. For instance, VAE or flow-based approaches could be cited in this section, plus any approach that seek to structure and control changes directly from manipulation of the latent representation.
> >    - Response: We rewrote the related work section and included your suggested VAE-based an flow-based methods as well as more recent works.
> >
> > Major C2: Fig2. vs Fig.3 are a bit misleading since the ranker model is acting on only 2 images while the "rank head" is taking 4.
> >    - Response: We updated Fig.2 and the loss to make it consistent and more clear.
> >
> > Major C3: section 3.4 and 3.5:  notation does not really simplify equations but rather gives rise to some obfuscation.
> >    - Response: We revised the equations accordingly.
> >
> > Major C4: In equation 5a, it does not seem possible to sample on $r$, since in fact, it depends on $(x,y)$ and  and should be written $r(x,y)$ for clarity.
> >    - Response: Note that $r$ is the relative attribute label of the pair of images $(\mathbf{x,y})$. We use the joint distribution to sample a triplet $(\mathbf{x,y},r)$, like the way defining the joint distribution of data and label $p(x,y)$.
> >
> > Minor C1: The "rival" wording used here and there from the abstract of the paper is not really introduced and self-supported.
> >    - Response: "rival" means adversarial. We use it to distinguish it from adversarial training in the community. We explain it in the revised paper to avoid confusion.
> >
> > Minor C2: In abstract: "real image preferences" was not clear for me before I see in Fig.2.
> >    - Response: we changed "real image preferences" into "preferences over pairs of real images" to make it more clear.
> >
> > Minor C4: In section 3.1, explaining how RCGAN uses its ranker is a bit confusing and should be better explained in the related work section.
> >   - Response: To avoid confusion, we removed the description of RCGAN in section 3.1 and included more details in the related work.
> >
> > Minor C5: In equation 5a, it does not seem possible to sample on $r$, since in fact, it depends on $(x,y)$ and should be written $r(x,y)$ for clarity.
> >    - Response: We don't agree with the reviewer. $r$ is the relative attribute label of the pair of images $(\mathbf{x,y})$. We use the joint distribution to sample a triplet $(\mathbf{x,y},r)$, like the way defining the joint distribution of data and label $p(x,y)$.
> >
> > Other minor comments: We revised them accordingly in our updated paper.
> >
> > Reference:
> >    * [1] Odena A. Semi-supervised learning with generative adversarial networks. 2016.
> >    * [2] Mao X, Li Q, Xie H, Lau RY, Wang Z, Paul Smolley S. Least squares generative adversarial networks. ICCV, 2017.
> >    * [3] Li, X., Lin, C., Li, R., Wang, C. \& Guerin, F., Latent Space Factorisation and Manipulation via Matrix Subspace Projection, ICML, 2020.

---

### Official Review · AnonReviewer3 · 2020-10-28
**A nice work if any theoretical analysis can be given**

**Rating:** 6
**Confidence:** 4

**Review:**

Summary: This paper proposes to improve the fine-grained image-to-image translation by utilizing an adversarial ranking framework. The new ranker helps the generator to have a better fine-grained control on the translation results. Experiment results exhibit that the proposed method achieves state-of-the-art results on several image translation tasks.


Major issues:
-  Although the intuition of the adversarial ranker makes sense, it will be better to give some theoretical analysis about when and how the optimal state of generator and ranker will achieve.
- Does the training scheme occur convergence problem? Because the model has three different networks and there has no explicit connection between ranker network and discriminator network.
- The experiments are mainly conducted on face datasets. It remains some questions about the generalization of the model to other non-face image translation tasks.
- There lacks the ablation study analyzing the proposed ranker module.

Minor issues
- How about the computation time increased by the ranker module?
- How do you choose the weighting factor for the networks?

---

> ### Author Response · Authors · 2020-11-16
> **We empirically analyzed the convergence of our TRIP and added more ablation studies as well as more visual comparisons**
>
> Thank you very much for your comments. Below is a report for how we address your concern.
>
> Issues 1: Although the intuition of the adversarial ranker makes sense, it will be better to give some theoretical analysis about when and how the optimal state of generator and ranker will achieve.
>    - Response: Note that the TRIP has a Nash Equilibrium when the generator $G$ produces the realistic image that changes the specific attributes over the input image as desired, and the ranker $R$ cannot distinguish the real image pairs from the generated images pairs in term of their attribute discrepancy.
>
>    However, as also discussed in the literature [1], it is still an open problem how an adversarial ranking process converges to such an equilibrium. To justify our claim, we empirically demonstrate the training convergence of our model. The ranker and the generator indeed converge to their optimal state as desired. We plot the distribution of the prediction for real image pairs and generated image pairs using the optimal ranker and the results are consistent with our analysis.
>
> Issues 2: Does the training scheme occur convergence problem? Because the model has three different networks and there has no explicit connection between the ranker network and discriminator network.
>    - Response: Our TRIP can converge well like other adversarial training based methods. There is no discriminator network in our model but only a ranker network, which contains rank head for ranking preferences and GAN head for image quality following a shared feature layer. We updated Fig.2 for better clarification. The ranker and the generator are trained against each other until convergence as shown in Fig. 11 in the appendix.
>
> Issues 3: The experiments are mainly conducted on face datasets. It remains some questions about the generalization of the model to other non-face image translation tasks.
>    - Response: In terms of the non-face image translation tasks, we applied our model to the UT Zappos50K dataset [1]. Our TRIP also achieves good performance on translating shoe images to edge images in a fine-grained manner.
>
> Issues 4: There lacks the ablation study analyzing the proposed ranker module.
>    - Response: We did the ablation study about the rank head loss and the GAN head loss shown in appendix A (originally submitted as the supplementary). The ranker module indeed facilitates the generator to learn the mapping of the translation
>
> Minor Issues:
> 	1) How about the computation time increased by the ranker module? 2) How do you choose the weighting factor for the networks?
>    - Response: 1) As clarified, we did not have two critic networks -- discriminator and ranker, but only a ranker network. As the ranker takes a pair of images as input, the computation time is doubled when passing the feature layer, which marginally increases the computation. 2) The tailor-designed weighting factors in our model are $\lambda$ and $\lambda_{g}$. We made a coarsen hyperparameter searching for these two factors by setting $0.5, 1, 5$, respectively. As the face images of LFW dataset have lower resolution than that of CelebA-HQ, we set up a larger $\lambda_{g}$ in order to maintain a good quality. Other weighting factors are set up following RelGAN.
>
> Reference
>    * [1] Zhu, J. Y., Park, T., Isola, P., & Efros, A. A.. Unpaired image-to-image translation using cycle-consistent adversarial networks. ICCV. 2017

---

### Official Review · AnonReviewer2 · 2020-10-29
**Incremental advances to a key module for fine-grained image-to-image translation.**

**Rating:** 5
**Confidence:** 4

**Review:**

This paper proposes to construct rival preference in the ranker to evoke the adversarial training between the ranker and the generator, leading to a better fine-grained control over the interested attribute in image-to-image translation task.

With tailor-designed loss functions, the ranker is promoted to pay attention on the effective information related to the target attribute and avoid providing biased prediction for unrealistic image pairs. Through the adversarial process between the generator and the ranker, the generator can improve the ability of generating desired attribute. Besides using the ranker to improve the ability of modeling desired attribute in the generator, a discriminator is adopted to improve the quality of generated images further. The design of proposed ranker is elegant and experimental results are promising.

This paper is well organized and clearly written.

Strengths:
1.	Proposed techniques are intuitive and well-motivated.
2.	Both qualitative results and quantitative comparisons demonstrate the superiority of proposed method compared with other methods.

Weaknesses:
1.	The main contribution of this paper is introducing adversarial learning process between the generator and the ranker. The innovation of this paper is concerned.
2.	Quality of generated images by proposed method is limited. While good continuous control is achieved, the realism of generated results showed in paper and supplemental material is limited.
3.	Visual comparisons and ablation study are insufficient.

Comments/Questions:
1.	Could you elaborate more on why proposed method achieves better fine-grained control over the interested attribute? Was it crucial to change the formular of ranker’s loss function from classification to regression?
2.	Could you provide more visual comparisons between the proposed method and prior works?
3.	There are also some other works focusing on the semantic face editing and they show the ability to achieve continuous control over different attributes, like [1]. Could you elaborate the difference between your work and these papers?
4.	Statements in Section 4.2 are somewhat redundant.

Minor:
1.	Missing proper expression for the third face image in Figure 2.
2.	Missing close parenthesis at the bottom of Page 4.
3.	Inconsistent statement and reference for Celeb Faces Attributes Dataset in experiment section.

[1] Shen, Yujun and Gu, Jinjin and Tang, Xiaoou and Zhou, Bolei. “Interpreting the Latent Space of GANs for Semantic Face Editing”, In CVPR, 2020. https://dblp.org/rec/conf/cvpr/ShenGTZ20

---

> ### Author Response · Authors · 2020-11-16
> **More visual comparisons and ablation study are added**
>
> Thank you very much for your comments. Below is a report for how we address your concern.
>
> Weakness 1:	The main contribution of this paper is introducing adversarial learning process between the generator and the ranker. The innovation of this paper is concerned.
>    - Response: This is the first work to formulate facial attribute transfer as an adversarial ranking problem, which achieves the state-of-art smooth facial attribute transfer.
>
> Weakness 2 \& Q2: Quality of generated images by proposed method is limited. While good continuous control is achieved, the realism of generated results showed in paper and supplemental material is limited.
>    - Response: We clarify our contribution in terms of image quality improvement more clearly in the revision. The desired high-quality images in our context are two folds: first, the generated images look as realistic as training images; second, the generated images are only modified in terms of the specific attributes. Our method achieves higher-quality images generation than all the baselines. Especially, our results are superior over current ranking-based methods, i.e., RCGAN (Fig. 5 \& Table 1).
>
> Weakness 3: 	Visual comparisons and ablation study are insufficient.
>    - Response: We include more visual comparison and more ablation study in the appendix. The added experiment is consistent with our analysis, which further demonstrates the superiority of the introduction of adversarial ranking for modeling facial attribute transfer.
>
> Q1:	Could you elaborate more on why proposed method achieves better fine-grained control over the interested attribute? Was it crucial to change the formula of ranker’s loss function from classification to regression?
>    - Response: The essential that our TRIP can achieve the SOTA results lies in the introduction of the generated images into the whole training process of the critic. Previous methods capture the subtle difference over attribute using a classification model or a ranking model and count on the learned attribute model to generalize the learned attribute preference to the unseen generated images through interpolation. As for our TRIP, we introduce the generated image into the training process of the attribute model, i.e., the ranker by formulating it into an adversarial ranking process. Since our ranker (the attribute model) can critic the generated image during its whole training process, it no doubt can generalize to generated images to ensure sufficient fine-grained control over the target attribute. See our general response to all reviewers for more details.
> The essential of this problem lies in how to ensure the generalization performance of the attribute model, i.e., ranker, on the generated images. Compared with a classification loss, adopted a regression loss for the ranker can enhance its generalization performance but it still by no means to ensure good generalization of the unseen generated images.
>
> Q3: There are also some other works focusing on the semantic face editing and they show the ability to achieve continuous control over different attributes, like [1]. Could you elaborate the difference between your work and these papers?
>    - Response: We rewrite the related work section so as to incorporate the recently published works.
>
> Q4:	Statements in Section 4.2 are somewhat redundant.
>    - Response: We understand the reviewer's concern that the results in Section 4.2 seem trivial. However, as we introduce an adversarial ranking process, the ranker's optimization becomes more complex. Actually, the interpolation discriminator introduced in RelGAN for the fine-grained generation fails to converge as desired due to the adversarial training. Our ranker instead keeps its output aligned well to the relative change in the pair of images, thus promoting a better generation.
> In addition, Section 4.2 justifies why we can apply our ranker to evaluate the subtle differences of the fine-grained synthetic image pairs, generated by different methods.
>
> Minor Comments: 1) Missing proper expression for the third face image in Figure 2. 2) Missing close parenthesis at the bottom of Page 4. 3) Inconsistent statement and reference for Celeb Faces Attributes Dataset in experiment section.
>    - Response: We revised the paper accordingly.

---

### Author Response · Authors · 2020-11-16
**To all reviewers: elaborate why our proposed TRIP achieves better fine-grained control over the interested attribute**

The essential that our TRIP can achieve the SOTA results lies in **the introduction of the generated images into to training of the critic.** As we all know, a good generator requires a good critic. To ensure good control over the target attribute, the critic should transfer the signal about the subtle difference over the target attribute to the generator.

Previous methods model it as two sequential processes, namely, they capture the subtle difference over attribute using a classification model or a ranking model, and count on the learned attribute model to generalize learned attribute preference to the unseen generated images through interpolation. However, the learned attribute model never meets our expectation, since they haven't seen the generated image at all during its training.

As for our TRIP, we consider introducing the generated image into the training process of the attribute model, i.e., the ranker, which explicitly models the relative attributes. Since the supervision over the generated images is not accessible, we formulate the ranker into an adversarial ranking process using the constructed rival preference, following the adversarial training of Vanilla GAN. Consequently, our ranker (the attribute model) can critic the generated image during its whole training process, and it no doubt can generalize to generated images to ensure sufficient fine-grained control over the target attribute.

**We updated a new version of our TRIP. The revision is highlighted in red.**

---

### Author Response · Authors · 2020-11-25
**To all reviewers: achieve good transformation results on non-facial data from shoe images to edge images**

According to the suggestion from the reviewers, we applied our TRIP on a non-facial dataset to conduct a style transfer, namely, transferring shoe images to edge images. Our TRIP can achieve good performance on translating shoe images to edge images in a fine-grained manner (seen in the appendix F). Due to the limited time, we only report our TRIP’s results. We will add the results of other baselines later.

---

### Decision · Program_Chairs · 2021-01-07
**Final Decision**

**Decision:**

Reject

**Comment:**

All reviewers gave either borderline or negative scores; unfortunately, discussion was not lively, so scores remained the same. No reviewers voice strong support for acceptance, but acknowledge several merits of the work.